# Transcriptional Responses for Biosynthesis of Triterpenoids in Exogenous Inducers Treated Inonotus Hispidus Using RNA-Seq

**DOI:** 10.3390/molecules27238541

**Published:** 2022-12-04

**Authors:** Yonghong Huo, Dongchao Liu, Qin Yang, Changyan Sun, Zhanbin Wang, Dehai Li

**Affiliations:** 1School of Forestry, Northeast Forestry University, Harbin 150040, China; 2Department of Environmental Engineering, School of Materials Science and Chemical Engineering, Harbin University of Science and Technology, Harbin 150040, China

**Keywords:** RNA-Seq, *Inonotus hispidus*, exogenous inducer, differential expressed genes

## Abstract

*Inonotus hispidus* is a traditional medicinal that grows in Northeast China and produces various economically important compounds, including polysaccharide compounds and terpenoids; triterpenoid saponins is the main bioactive component. Our research group has found that the accumulation of triterpenoid was affected by exogenous inducers. Experimental results showed that treatment with methyl jasmonate (MeJA) and oleic acid significantly increased the triterpenoid content of *I. hispidus*. However, how exogenous inducers enhance production of secondary metabolites in *I. hispidus* is not well understood. In this study, metabolite changes were further investigated with UPLC-TOF/MS following exogenous inducer treatment. As a result, a total of eight types of triterpenoids in *I. hispidus* were identified. The RNA-seq analysis was used to evaluate the effects of exogenous inducers on the expression of triterpenoid-synthesis-related genes in *I. hispidus* in liquid fermentation. This study is the first exploration to profile the transcriptome of *I. hispidus* after adding exogenous inducers; the generated data and gene will facilitate further molecular studies on the physiology and metabolism in this fungi. By comparative transcriptomic analysis, a series of candidate genes involved in the biosynthetic pathway of triterpenoids are identified, providing new insights into their biosynthesis at the transcriptome level.

## 1. Introduction

*Inonotus hispidus*, also known as *fusarosa*, belongs to *Basidiomycetes*, *Hymenomycetes*, *Polyporales*, which is a kind of rare medicinal fungus [1]. Every year from June to September, the subbody grows in large numbers, mainly parasitic, on Manchurian Ash, elm, poplar, mulberry and so on; it is distributed in Beijing, Hebei, Inner Mongolia, northeast China, Xinjiang and other places. Triterpenoids are considered as the main bioactive constituent of *I. hispidus*. Modern pharmacology research has proved its extensive pharmacological effects, including anti-tumor, anti-oxidation, anti-inflammation, anti-virus, hypidemia, hypoglycemia and liver protection [2,3]. Although it is a traditional Chinese medicine, *I. hispidus* is playing its own unique medicinal value, and the molecular mechanisms underlying the accumulation of bioactive triterpenoids are poorly understood. With the vigorous development of health industry, it has become a trend to vigorously develop *I. hispidus* in the field of edibles and medicinals to achieve industrialization development. With demand increasing, it is important to take effective measures to increase the accumulation of triterpenoids in *I. hispidus*.

Progress has been achieved over the years in enhancing the triterpenoid content either by changing the medium composition, adjusting the fermentation parameters, or adding elicitors [4]. Inducers can act as a special chemical signal in liquid fermentation, which can effectively selectively induce the expression of specific genes in the fungi metabolic pathway [5,6]. This can also be recognized by receptors located on the surface of the plasma membrane or endomembrane. Receptors are activated and then activate their effectors, such as ion channels, GTP binding proteins, protein kinases and oxidative bursts [7]. MeJA has been widely used to stimulate fungi and plant stress responses, promote the expression of related genes in secondary metabolic pathways, and increase the production of secondary metabolites such as phytoalexins, alkaloids, flavonoids, phenolic acids and terpenes [8,9]. Research shows that when 150 μmol/L MeJA was added to the submerged fermentation of *Inonotus baumii*, the triterpenoid content increased by 4.05 times compared to the control group [10]. Oleic acid is an unsaturated fatty acid that exists in animals and plants [11]. Oleic acid has been used as a stimulatory agent to increase the production of triterpenes and promote the growth of mycelia [12]. Zhou et al. [13] showed that oleic acid as an inducer increased the triterpenoid content of *I. hispidus* by 143.74%. Notwithstanding, the physiological processes and secondary metabolism of *I. hispidus* treated with the exogenous inducer remains largely unknown.

An increasing number of studies have focused on the transcriptome and genome levels with an understanding of plant secondary metabolic pathways including *Cordyceps* [14], *Ophiocordyceps sinensis* [15] and white *Hypsizygus marmoreus* [16], among others. However, there has still not been a genome-wide expression study of *I. hispidus*. Therefore, the purpose of this study was to explore the effects of exogenous inducers on the triterpenoid contents of *I. hispidus* through comparative transcriptome analysis, differentially expressed genes (DEGs) related to triterpenoids accumulation in *I. hispidus* induced by MeJA, oleic acid and formulation. We further explored the transcriptomic data combined with triterpenoid accumulation and provided more possibilities for biotechnology to improve the medicinal value of *I. hispidus*.

## 2. Results

### 2.1. Effect of Inducers on Triterpenoid Content of I. hispidus

The triterpenoid content in *I. hispidus* treated with 50 μmol/L MeJA, 3% oleic acid, and a formulation (2% oleic acid and 100 μmol/L MeJA) increased significantly compared with that in the CK. Among them, the triterpenoid content in *I. hispidus* treated with the formulation was the highest. The content of mycelium and triterpenoid were 14.83 g/L and 157.46 mg/g, which were 33.72% and 145.53% higher than the blank control, respectively, as shown in Figure 1. This indicated that the combination of MeJA and oleic acid has a certain synergistic effect on promoting triterpenoid synthesis in *I. hispidus*. This may be because oleic acid acts as an auxiliary carbon source to promote mycelial growth in the early stage of the mycelial fermentation of *I. hispidus*, and acts as an elicitor in the later stage to promote the genes of key enzymes (SQS, CYP51, etc.) in the triterpenoid synthesis pathway expression. MeJA can be used as a second messenger in secondary metabolic synthesis in the late stage of mycelial fermentation; conversely, it can also increase the concentration of reactive oxygen species in the mycelium and promote the expression of key enzymes in the triterpenoid synthesis pathway.

### 2.2. Identification of Triterpenoid of I. hispidus

UPLC-TOF-MS/MS was used to qualitatively analyze the mycelial triterpenoids of *I. hispidus* after the compound induction of MeJA and oleic acid and obtain the total ion flow diagram, as shown in Figure 2. UPLC-TOF-MS/MS technology was used to collect the data, and Compound Discover 3.1 software was used to automatically identify and analyze the triterpenoid chemical constituents of *I. hispidus*. A total of eight triterpenoids were identified of *I. hispidus*, all of which were pentacyclic triterpenes. By sorting out the data provided by mass spectrometry—combined with the elution sequence, retention time, molecular weight, mass spectrometry information, etc.—of each compound, the possible molecular formula and preliminary identification results are shown in Table 1. It has been reported in the literature that tetracyclic triterpenoids and pentacyclic triterpenoids are the most commonly found triterpenoids. Among them, medicinal fungi and plants contain more species of pentacyclic triterpenoids, and their structures are divided into four types. Classes: Ursane, oleanane, lupinane and suberane [17]. Studies have found that maslinic acid has a good ability to resist colon cancer, breast cancer and other tumors [18]. Oleanolic acid can promote the up-regulation of gene expression of key enzymes of reproductive differentiation in mice, and promote the rapid differentiation of various embryonic stem cells into Germ cells, which facilitate the synthesis of germ cells and the secretion of estrogen [19]. This experimental study found that most of the triterpenoid monomers of *I. hispidus* are oleanane-type, and oleanane-type triterpenoid acids have pharmacological effects such as hypoglycemic, hypolipidemic, antitumor, etc. [20].

### 2.3. RNA-Seq and De Novo Assembly of I. hispidus Reference Transcriptome

RNA-seq on *I. hispidus* with different treatments was performed using the Illumina HiSeq 4000 sequencing platform, which resulted in four samples named CK and MeJA(S_1_), Oleic acid (S_2_), MeJA and Oleic acid (S_3_) with a total of 12 sequencing libraries. After filtering out the low-quality reads, the results in Table 2 show that a total of 82,680,165 clean reads in CK, 82,271,188 clean reads in the S_1_, 91,676,834 clean reads in the S_2_ and 96,148,730 clean reads in S_3_ were obtained. The total clean base numbers of 24.64 G (CK), 24.51 G(S_1_), 27.33 G(S_2_) and 28.62 G(S_3_) were obtained, respectively. The percentage of Q30 was 92.98% or higher, and the percentage of GC was approximately 52%.

### 2.4. Functional Annotation and Classification

In order to obtain comprehensive gene function information, all the assembled unigenes were successfully annotated using eight public databases (Nr, Nt, Pfam, KOG, COG, Swiss-prot, KEGG, GO). As shown in Table 4, out of 19,429 unigenes, the annotation success rates of the unigenes were: 6537 in COG (33.65%), 6884 in GO (35.76%), 6640 in KEGG (33.5%), 11,968 in KOG (62.17%), 12,430 in Pfam (63.98%), 8355 in Swiss-prot (43%), 17,914 in eggNOG (92.2%) and 12,754 in NR (65.60%). The functions of the predicted unigenes were classified in KOG, GO and KEGG.

The species distribution of the Nr annotation showed that 36.22% (4606) of the unigenes had the highest homology to genes from *Sanghuangporus baumii*, 13.72% (1744) to *Fomitporia mediterranea*, 2.62% (333) to *Basidiobolus meristosporus* and 2.56% (325) to *Spizellomyces punctatus*. (Figure 3).

KOG results show a total of 13,272 unigenes divided into 25 categories (Figure 4a). Among these genes, the cluster “General function prediction only” cluster represented the largest group (R) (1797, 15.02%), followed by posttranslational modification, protein turnover, chaperones (O) (1670, 13.95%), translation, ribosomal structure and biogenesis (J) (1080, 9.02%), and signal transduction mechanisms(T) (1077, 9%). In addition, 418 (3.49%) unigenes were associated with secondary metabolite biosynthesis, transport and catabolism, and 105 were associated with defense mechanism unigenes (V).

According to sequence homology, 39,274 unigenes were classified into three GO categories: biological process (14511, 36.95%), cellular component (16952, 43.16%), and molecular function (7811, 19.89%) (Figure 4b). The most popular terms in the biological process category were “metabolic processes (3868)”, “cellular processes (3778)” and “single-organism process (2406)”. In the cellular component category, the most common terms were “cell (3831)”, “cell parts (3811)”, and “organelle (2758)”. In the molecular function category, the main terms were “catalytic activity (3505)” and “binding (3124)”, suggesting that this study may provide clues to understanding genes involved in secondary metabolite synthesis pathways. Transcripts associated with binding GO terms were more common in the molecular functional category.

Comparing Unigenes with standard metabolic pathways in the KEGG database to better understand the biological pathways of *I. hispidus* leaves (Figure 4c), the 6640 Unigenes are divided into 4 KEGG categories and 117 sub categories: “metabolism”, “genetic information processing”, “environmental information processing” and “cellular processes”. In the KEGG secondary metabolic pathways, most unigenes were classified into “Biosynthesis of antibiotics” (606 unigenes, ko01130), “Ribosome” (499 unigenes, ko03010), “Carbon metabolism” (364 unigenes, ko01200) and “Biosynthesis of amino acids” (324 unigenes, ko01230).

### 2.5. Identification and Analysis of DEGs

A total of 241 upregulated DEGs and 112 downregulated DEGs were identified in the group ‘S1 vs. CK’, 635 up regulated DEGs and 443 down regulated DEGs were identified in the group ‘S2 vs. CK’ and 529 up regulated DEGs and 364 down regulated DEGs were identified in the group ‘S3 vs. CK’ (Figure 5a–c). A total of 162 core DEGs (co-upregulated or co-downregulated in all three exogenous-inducer-treated samples) were obtained in the three comparison groups (The 162 core DEGs can be searched in Appendix A). Furthermore, overlapping studies found that there were 150, 296 and 126 unique genes in the S1, S2 and S3 (Figure 5d).

### 2.6. Enrichment Analysis of DEGs

To compare the unigenes from different inducer-treated *I. hispidus*, a Venn diagram was constructed. The results showed that 6777 unigenes were shared by four sample. A total of 54, 61, 44 and 43 unigenes were specific to CK, MeJA, Oleic acid and formulation treated *I. hispidus*, respectively, with the MeJA treated samples having the highest number of unique unigenes. (Figure 6a).

KEGG pathway analysis of all DEGs was performed to characterize the complex biological behaviors. The enriched pathways are presented in Figure 5 and reflect the preferential biological functions of samples from different inducer treated samples. In ‘S1 vs. CK’, genes involved in “Ribosome”, “DNA replication” and “Cell cycle—yeast” were overexpressed (Figure 6b). The KEGG pathway enrichment of all DEGs indicated that parts of the unigene enrichment characteristics were similar between the ‘S2 vs. CK’ and ‘S3 vs. CK’ comparisons, with genes involved in “Ribosome”, “Peroxisome” and “ABC transporters” over-expressed (Figure 6c,d). The major GO enrichment terms of the 162 core DEGs were shown in Figure 7a, including “translation”, “ribosome” and “structural constituent of ribosome”, etc. The KEGG pathway enrichment analysis showed that most of the core DEGs were significantly enriched in the ribosome pathways (Figure 7b). Moreover, the hierarchical cluster analysis was carried out in the form of a heatmap, which could represent the expression levels of these transcripts under specific elicitation. We found that these genes contain the hydroxylethylglycaryl-CoA synthase (ID: c16013.graph_c0) related to triterpenoid synthesis and antioxidant enzymes in the defense system (ID: c10077.graph_c0). Interestingly, the expression levels in the samples are relatively high. These data assist in the investigation of specific functions, processes, and pathways, and can also help in the identification of genes related to triterpenoid biosynthesis in *I. hispidus*.

### 2.7. Putative Genes Involved in Triterpenoid Biosynthesis

Triterpenoids are widely found in nature [21]. In plants, sesquiterpenoids are typically synthesized via mevalonate (MVA) and 2-methyl-d-erythritol 4-phosphate (MEP) biosynthetic pathways, as shown in Figure 8 [22]. Although the MVA and MEP pathways are located in different intracellular regions, the two pathways are not separated. Additionally, there may be unknown interference between certain devices. It is generally believed that triterpenes and sesquiterpenes are synthesized through the MVA pathway, while monoterpenes and diterpenes are synthesized through the MEP pathway [23]. IPP and DMAPP are catalyzed by the sequential conversion of geranyl diphosphate synthase (GPPS), clopyridine diphosphate synthase (FPPS), squalene synthase (SS) and squalene epodase (SE). It is an important intermediate for the oxidation of 2,3 squalene [24,25].

After annotating 117 pathways using the KEGG database, 29, 13 and 7 unigenes were found to be associated with Terpenoid backbone biosynthesis (ko00900), Ubiquinone and other terpenoid-quinone biosynthesis(ko00130) and Sesquiterpenoid and triterpenoid biosynthesis(ko00909) pathways, respectively. Comparing the NR database, a total of 29 unigenes that may be involved in the triterpene synthesis pathway of *I. hispidus* were found (Table 5). The related unigenes can be searched in Appendix A.

### 2.8. Analysis of CYP450 Defense Genes

CYP450 play important roles in plant defense through their involvement in phytoalexin biosynthesis, hormone metabolism and the biosynthesis of some other secondary metabolites [26,27]. However, the systematic identification of CYP450 has not been reported in *I. hispidus*. By searching the annotation results of the transcriptome database of *I. hispidus*, we found that a total of 90 unigenes were annotated as CYP450, of which 66 CYP450 unigenes were annotated into the “defense mechanisms”. We speculate that due to the addition of exogenous inducers, *I. hispidus* causes a defense response and promotes the synthesis of triterpenes. Most CYP450 annotated to the “defense mechanisms” also illustrate this view. After the addition of the inducer, 1, 10 and 9 CYP450 genes were up-regulated in S_1_, S_2_ and S_3_ groups, respectively. Therefore, in this study, it can be speculated that the up regulation of CYP450 promotes the synthesis of triterpenoids, which will help to reveal its regulatory mechanism in the terpenoid synthesis pathway of *I. hispidus*.

In transcriptome sequencing data of *I. hispidus*, we found 2 SS, 6 se, 2 FPPS and other terpene-related genes. Cytochrome P450 plays an important role in the terpene biosynthesis. Monoterpenes, sesquiterpenes, and diterpenes can be substrates of P450 [28]. For example, Guo [29] showed that CYP76AH3 and CYP76AK1 are involved in the tanshinone biosynthesis. In this study, 66 CYP450 genes were found in the defense pathway; among them, 1, 10 and 9 upregulated genes were in the MeJA, oleic acid and formulation groups, respectively. Therefore, the accumulation of the triterpenoid content in *I. hispidus* might be due to exogenous inducer treatment, which activated and improved the CYP450 expression.

### 2.9. Analysis of Defensive Enzymatic Defense Genes

As a traditional Chinese medicinal fungus, *I. hispidus* has been studied for its antitumor [30], bacteriostatic [31] and other efficacy in the past. Elicitation is one of the most effective ways to increase the activity of plant compounds such as triterpenes [32]. The SOD, CAT and POD are important cytoprotective enzymes that eliminate reactive oxygen species in plants. Its activity may reflect plant adaptation to abiotic stress. [33,34]. Exogenous inducers can affect the secondary metabolism of plants by altering the metabolism of reactive oxygen species (ROS), resulting in an increase in secondary metabolites [35]. Naeem et al. [36] showed that salicylic acid can induce reactive oxygen species bursts, up-regulate artemisinin gene expression and increase artemisinin production. In the transcriptome sequencing data of *I. hispidus*, 34 genes annotated as POD, 7 genes annotated as CAT and 5 genes annotated as SOD. However, we found that only three POD genes were upregulated, and the others are normal genes. MeJA elicits rapid defense response changes in metabolic processes dramatically toward the energy supply [37]. Therefore, under the MeJA treatment of *I. hispidus*, the increase in the antioxidant enzyme activity was promoted through POD upregulation to regulate the biosynthesis of secondary metabolites, such as triterpenoid. However, how oleic acid stimulates the metabolism needs further exploration.

## 3. Discussion

Even though *I. hispidus* has great economic and pharmacological utility, there have been no transcriptomic databases for this plant constructed to date, and no studies have systematically identified genes involved in triterpenoid biosynthesis. Our Illumina HiSeq 4000 sequencing provides the first transcriptome dataset, which provides novel insights into the mechanism of abiotic stress responses in *I. hispidus*. This study would be helpful to understand the mechanism of triterpenoid biosynthesis in *I. hispidus* at the molecular level. Creating a cDNA library from the *I. hispidus* and performing RNA-seq and DEGs provided a very efficient means for identifying the genes associated with known enzymes involved in the biosynthesis of secondary metabolites and for providing candidate genes that could be associated with currently unknown steps in the pathway. Based on bioinformatic analysis, all possible enzymes involved in the triterpenoid biosynthetic pathway of *I. hispidus* were identified. Among the 117 pathways were 3 pathways related to triterpenoid biosynthesis. These unigenes were matched to the function genes that were involved in the metabolic pathway of triterpenoid biosynthesis. Additionally, a total of 66 putative cytochrome P450(CYP450) were selected as the candidates of triterpenoid saponin modifiers. Our study shall greatly help further molecular cloning and functional identification of triterpenoid biosynthesis genes in *I. hispidus*. Based on this assumption, CYP450 with unknown functions are highlighted in our analysis because they are most likely the committed enzymes for those unclear steps in the triterpenoid biosynthesis pathway. Given the very low growth and biomass accumulation of the plant, it is expected that a triterpenoid pathway may be engineered in model species such as yeast to produce rare triterpenoid saponins from *I. hispidus* in a large-scale.

## 4. Materials and Methods

### 4.1. Experimental Materials and Design

The strain of *Inonotus hispidus* (IH-69) was sourced from the Key Laboratory Forest Food Resources Utilization of Heilongjiang Province. The strain was maintained on potato-agar-dextrose slants and cultured at 28 °C for 7 days. According to previous methods, culture media containing the following components (g/L) were prepared: fructose (40), Beef paste (2), KH2PO4(1), MgSO4.7H2O (0.5). Three mycelial pieces with a diameter of 12 mm were intercepted and inoculated in a liquid medium. Accurately weigh 50 μmoL MeJA with Twean-20 (0.2%) as cosolvent, and prepare the solution with 0.0025 mol/L concentration of sterile water at a constant volume of 100 mL. Then filter the solution with a 0.2μm filter membrane for sterilization, so that the final concentration of MeJA added to the medium is 50 μmol/L. On the sixth day of fermentation, it was added as the first experiment. A total of 3% oleic acid was added on the 0 day of fermentation as the second experiment, 2% oleic acid was added on the 0 day of fermentation and 100 μmol/L MeJA was added on the 6 Day of fermentation as the third group. The culture cycle of each group was 10 days. The group without an inducer was used as the blank control. The samples were snap-frozen in liquid nitrogen after a quick rinse with sterile water and stored at −80 °C for subsequent analysis. All determinations were carried out in triplicate.

### 4.2. Measurement of Triterpenoid Content in I. hispidus

Triterpenoids were extracted and quantified according to previous reports [38]. To extract the triterpenoids of *I. hispidus*, 100 mg dried mycelia were solubilized with 2 mL of 72% ethanol (*v/v*) and then placed in an ultrasonic chamber for 30 min at 210 W of ultrasound (40 kHz). Afterwards, the mycelia were removed by centrifugation at 4000× *g* for 10 min, and the supernatant was dried at 100 °C in a thermostatic water bath. To quantify the amount of triterpenoid, the residues were resuspended in 5% vanillin-acetic acid and perchloric acid, incubated at 70 °C for 15 min and cooled rapidly. Then, 4 mL of ethyl acetate was added, and the absorbance at 551 nm was measured with a spectrophotometer. The triterpenoid content was calculated as the betulin equivalent from a standard curve using betulin.

### 4.3. Metabolite Extraction Analyses in I. hispidus

Metabolites were extracted from three replicates for each MeJA and oleic acid treatment. A total of 100 mg samples were weighed, and the metabolites were extracted using a 1000 μL methanol: water (4:1, *v/v*) solution, and 30 μL Oleanolic Acid (0.3 mg/mL) was added as an internal standard. The supernatant was obtained by ultrasonic extraction at 210 W for 30 min, centrifugation at 4000 r/min for 10 min, and the supernatant (200 μL) was carefully transferred to sample vials for ultraperformance liquid chromatography/tandem mass spectrometry (UPLC–MS/MS) analysis. Metabolites were profiled using a UPLC-Triple-time-of-flight (TOF)-MS-based platform.

### 4.4. Library Preparation for Transcriptome Sequencing

Total RNA was extracted from control and exogenous-inducer-treated samples using the RNA prep Pure Plant Kit (Tiangen, China) based on the manufacturer’s manual. DNA was then removed using the RNA clean Kit (Tiangen, China). The quality and quantity of total RNA was assessed with the Agilent 2100 Bioanalyzer (Agilent Technologies, CA, USA). The library construction and RNA-Seq assay were performed by the Biomarker Biotechnology Corporation (Beijing, China). Poly-(A)-containing mRNA was purified from the total RNA using oligo (dT) magnetic beads and Oligotex mRNA kits (Qiagen, Hilden, Germany), following the manufacturer’s instructions. Fragmentation was carried out using divalent cations. Fragmentary RNAs were used as the template for first strand cDNA synthesis by a cDNA preparation kit. The cDNA was synthesized using RNase H and DNA polymerase I. Then, cDNAs were subjected to end repair, phosphorylation and ligation to sequencing adapters. Afterward, the products enriched by PCR amplification were purified through 2% agarose gelelectrophoresis and quantified by TBS380 (Picogreen). Finally, cDNA libraries were subsequently sequenced using an Illumina HiSeq 4000 platform.

### 4.5. Functional Annotation and Metabolic Pathway Analysis of Illumina HiSeq

The unigene sequences were compared with those in the National Center for Biotechnology Information non-redundant database (Nr) by using BLAST software with the E value set to ≤ 10^−5^, and the functional annotations of the proteins were obtained [39]. Moreover, the unigenes searched against different databases, including NCBI non redundant protein sequences (Nr), NCBI non-redundant nucleotide sequences (Nt), Protein primogenomic cluster (KOG/COG), Artificial annotated and reviewed protein sequence database (Swiss-Prot), Kyoto Encyclopedia of Genes and Genomes (KEGG), Gene Ontology (GO) and Pfam to classify the function and analyze the biological metabolic pathways of the unigenes [40].

### 4.6. Analysis of Differential Genes

Fragments per kilobase of transcript per million mapped reads (FPKM)is the number of reads per thousand base length of a gene compared in each million reads, it is a common method to estimate gene expression level in transcriptome sequencing data analysis. The FPKM values were used to evaluate the gene expression levels [41]. Taking the blank sample as a control, differentially expressed genes (DEGs) analysis in pair-wise comparisons was conducted using the DESeq2_EBSeq software, and the input data of the DEGs was based on the read counts [42]. The significant *p*-value was corrected. The corrected *p*-values were used as key indicators for DEG screening to obtain independent statistical parameters related to the expression of a large number of genes. In the screening process, the *p*-value was <0.05 and the fold change (FC) difference >1.5 was used as a screening criterion. Furthermore, the DEGs were then analyzed through GO and KEGG pathway enrichment analysis.

### 4.7. Statistical Analysis

Experimental data are given as the mean ± standard deviation with three replications. Charts were processed using Origin 9.0 software, and significance analyses were performed with SPSS 21.0 software. Different letters in the figures indicate a significant difference (*p* < 0.05). The differences between the amounts of mycelia are represented by A, B, C and D, and the differences between the amounts of triterpenoids are represented by a, b, c and d. All RNA-Seq analysis and mapping are carried out using Biomarker Biotechnology Corporation platform, the names and versions of all software used for RNA-seq analysis are detailed in Appendix A.

## Figures and Tables

**Figure 1 molecules-27-08541-f001:**
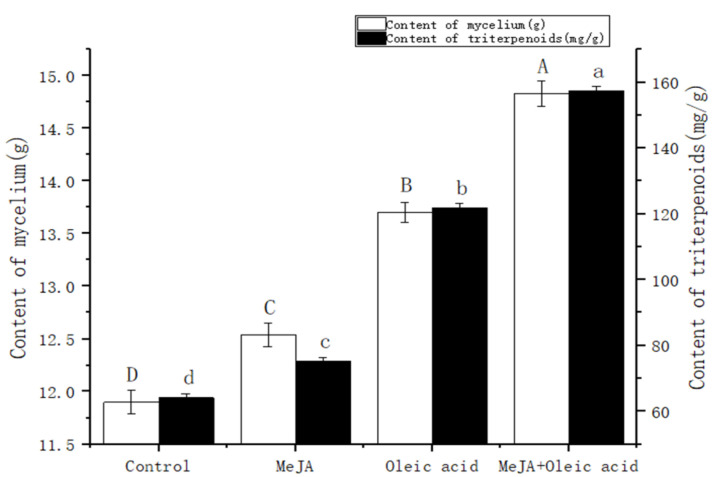
Triterpenoid content under different inducers. Different letters in the figures indicate a significant difference (*p* < 0.05). The differences between the amounts of mycelia are represented by A, B, C and D, and the differences between the amounts of triterpenoids are represented by a, b, c and d.

**Figure 2 molecules-27-08541-f002:**
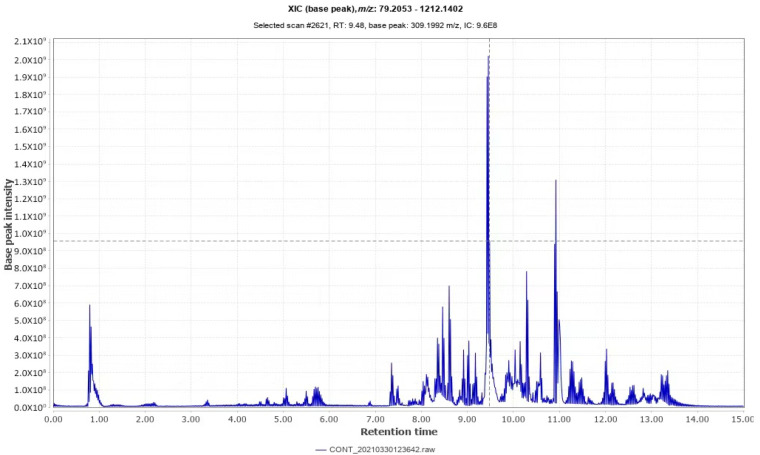
UPLC-TOF-MS/MS analysis of the TIC chromatogram of the triterpene extract of *Inonotus hispidus*.

**Figure 3 molecules-27-08541-f003:**
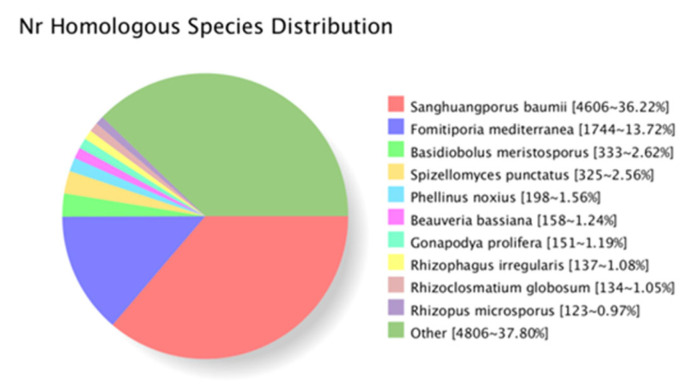
Distribution of unigene homologous species.

**Figure 4 molecules-27-08541-f004:**
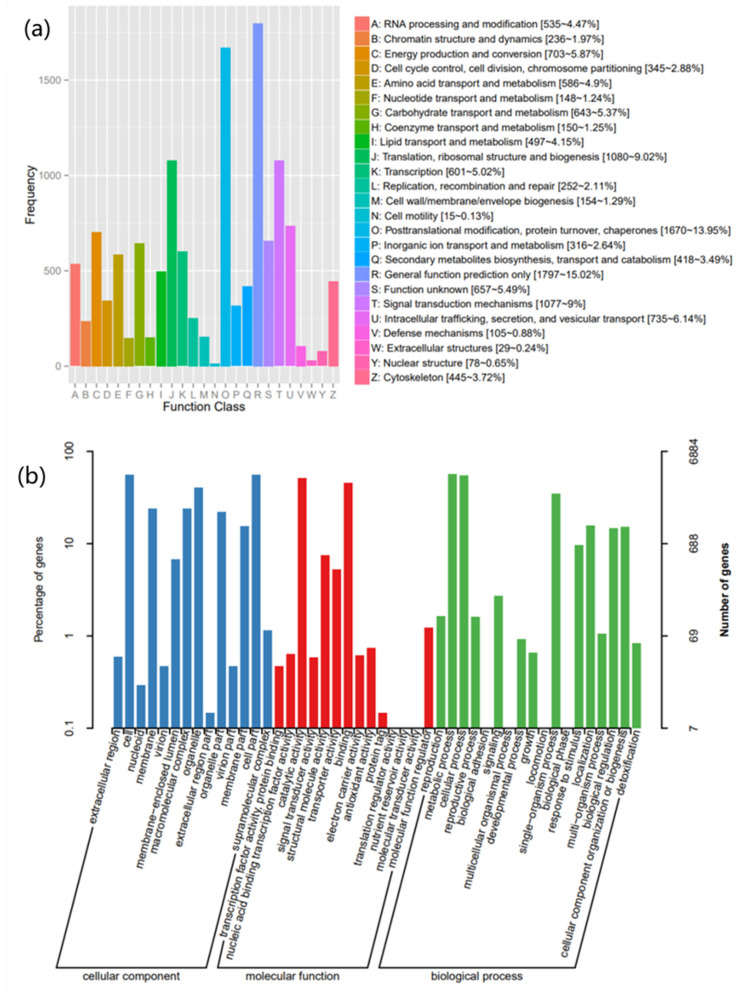
(**a**) KOG Function Classification of Consensus Sequence. (**b**) Gene ontology classifications of the assembled unisequences. (**c**) Functional classification and pathway assignment of assembled unigenes by KEGG. The assembled unigenes were classified into four main categories in KEGG classification. The *x*-axis indicates the number of unigenes in the category, and the *y*-axis indicates the KEGG classification.

**Figure 5 molecules-27-08541-f005:**
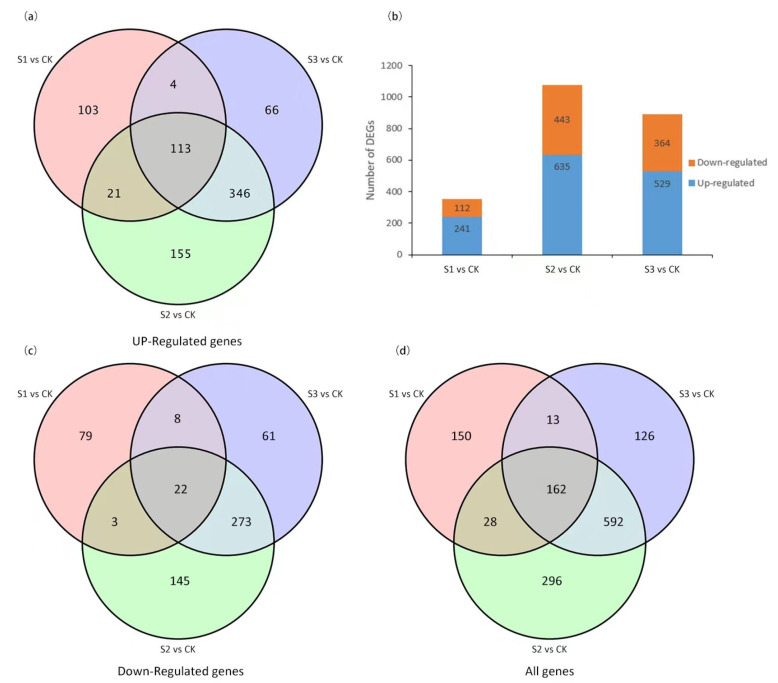
Venn diagrams of DEGs among different exogenous inducer. (**a**) Venn diagram showed the number of up-regulated genes in ‘S1 vs. CK’, ‘S2 vs. CK’ and ‘S3 vs. CK’ (**b**) The number of up-down regulated DEGs of S1 vs. CK, S2 vs. CK and S3 vs. CK. The up regulated DEGs are marked with blue, the down regulated DEGs are marked with orange-red (**c**) Venn diagram showed the number of down-regulated genes in ‘S1 vs. CK’, ‘S2 vs. CK’ and ‘S3 vs. CK’ (**d**) Venn diagram showed the number of all genes in ‘S1 vs. CK’, ‘S2 vs. CK’ and ‘S3 vs. CK’.

**Figure 6 molecules-27-08541-f006:**
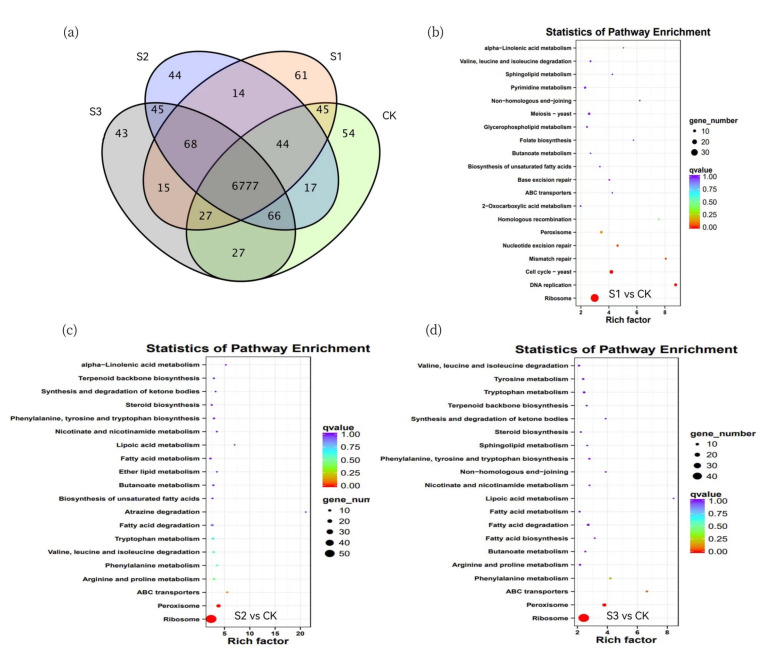
(**a**) Venn diagrams of *I. hispidus* treated with different inducers. (**b**–**d**) The KEGG pathway enrichment analysis of DEGs in *I. hispidus* of (**b**) MeJA, (**c**) Oleic acid and (**d**) formulation.

**Figure 7 molecules-27-08541-f007:**
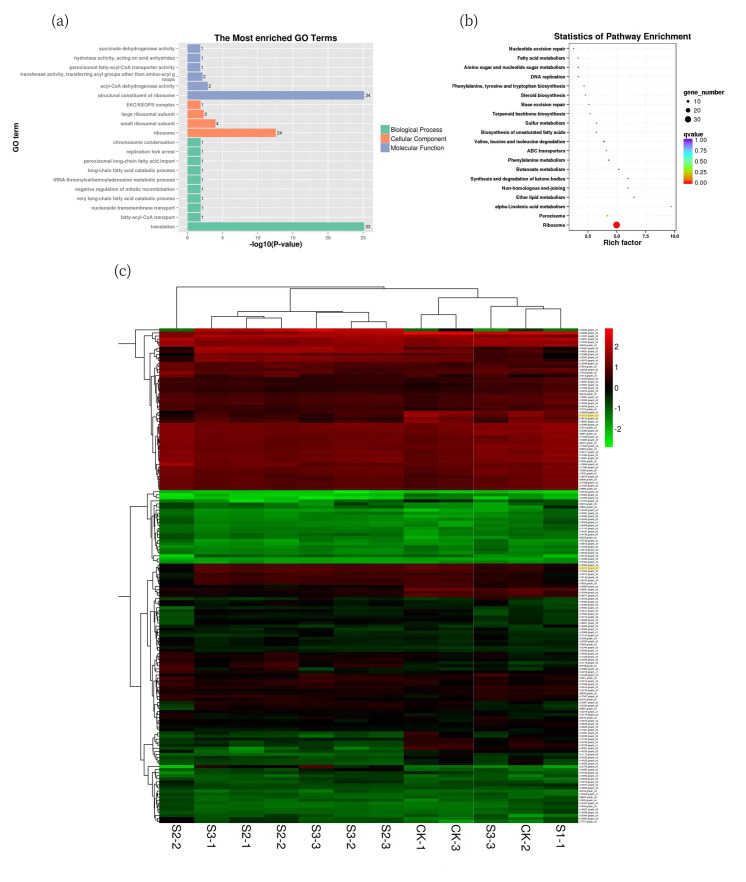
(**a**) GO Terms enriched of core DEGs. (**b**) KEGG enrichment analysis of core DEGs. (**c**) Hierarchical cluster heatmap of core DEGs, each row represents a gene, and each column represents a sample. The color changes from red to green according to the amount of expression.

**Figure 8 molecules-27-08541-f008:**
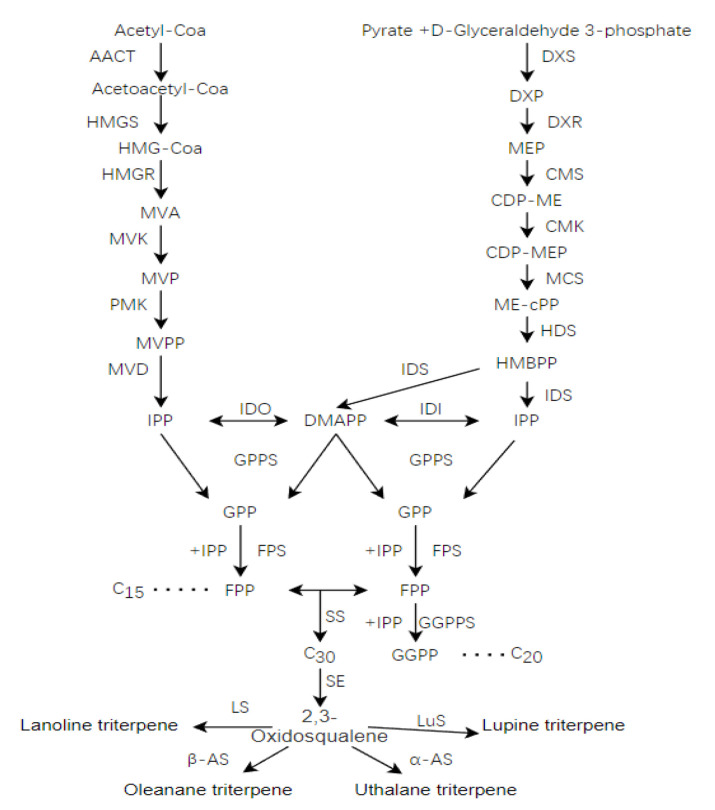
Biosynthetic pathway of terpenoids in *I. hispidus*.

**Table 1 molecules-27-08541-t001:** LC-MS preliminary analysis and identification of triterpenoids in *Inonotus hispidus*.

Number	Keep Time /min	Adduct	Theoretical Value (*m*/*z*)	Measured Value (*m*/*z*)	Molecular Formula	Error (ppm)	Inferred Compound
1	9.335	[M − H]^−^	488.3491	487.3379	C_30_H_48_O_5_	2.1	calyxagenin
2	9.482	[M − H]^−^	488.3491	487.3372	C_30_H_48_O_5_	−0.7	Asiatic acid
3	9.763	[M − H]^−^	486.3334	485.3315	C_30_H_46_O_5_	3.9	Abrisapogenol I
4	11.598	[M − H]^−^	472.3541	471.3237	C_30_H_48_O_4_	2.5	Maslinic acid.
5	11.779	[M − H]^−^	486.3697	485.3518	C_31_H_50_O_4_	0.78	Methyl Masridate
6	12.200	[M − H]^−^	484.3539	483.3307	C_30_H_48_O_3_	3.3	Ursolic acid.
7	12.948	[M − H]^−^	456.3591	455.3528	C_30_H_48_O_3_	−2.8	Oleanolic acid
8	13.241	[M − H]^−^	470.3742	469.3512	C_31_H_50_O_3_	4.2	methyl oleanate

**Table 2 molecules-27-08541-t002:** Summary of Illumina HiSeq 4000 sequencing data.

Sample	Clean Reads	Clean Bases	Q30	GC
CK-1	28,043,969	8.35 G	94.15	52.01
CK-2	27,100,025	8.07 G	94.32	51.96
CK-3	27,536,171	8.22 G	94.21	51.87
S_1_-1	30,910,974	9.21 G	94.21	51.92
S_1_-2	25,327,950	7.55 G	94.07	51.78
S_1_-3	26,032,264	7.75 G	94.31	51.91
S_2_-1	34,705,547	10.36 G	94.29	51.82
S_2_-2	27,883,532	8.30 G	94.60	51.87
S_2_-3	29,087,755	8.67 G	94.14	51.85
S_3_-1	32,483,363	9.66 G	94.09	51.81
S_3_-2	30,986,102	9.22 G	92.98	51.94
S_3_-3	32,679,265	9.74 G	94.17	51.91

*I. hispidus* clean reads were assembled from 12 libraries using Trinity software to obtain 300,423 transcripts. The average read length and N50 length were 3407.04 and 4385 bp, respectively (Table 3). By continuing to assemble these transcripts using Trinity, a total of 24,306 unigenes were spliced, with an average read length and N50 length of 1096.06 and 2687 bp, respectively. Among them, 9520 (39.17%) were 200–300 bp, 4738 (19.49%) were 300–500 bp, 3067 (12.62%) were 500–1 kb and 2784 (11.45%) were 1–2 kbp. A total of 4197 (17.27%) were longer than 2 kbps.

**Table 3 molecules-27-08541-t003:** Summary of splicing transcripts.

Length_Interval	200–300 bp	300–500 bp	500 bp–1 kbp	1 kbp–2 kbp	>2k bp	Total
Number of transcripts	10947	7551	12949	56309	212667	300423
Number of unigenes	9520	4738	3067	2784	4197	24306

**Table 4 molecules-27-08541-t004:** Function annotation of unigenes.

Database	Number of Unigenes	Percentage
Annotated in COG	6537	33.65%
Annotated in GO	6884	35.76%
Annotated in KEGG	6640	33.5%
Annotated in KOG	11,968	62.17%
Annotated in Pfam	12,430	63.98%
Annotated in Swiss-prot	8355	43%
Annotated in eggNOG	17,914	92.2%
Annotated in NR	12,754	65.60%
Total unigenes	19,429	100%

**Table 5 molecules-27-08541-t005:** Unigenes involved in Biosynthesis of Triterpenoid.

Gene	Gene Number	Gene ID
SS	2	c7497.graph_c0, c14961.graph_c0
SE	6	c16422.graph_c0, c15729.graph_c0,
		c24729.graph_c0, c20215.graph_c0
		c61.graph_c0, c19680.graph_c0
AACT	4	c8971.graph_c0, c16837.graph_c0
		c7417.graph_c0, c18532.graph_c0
HMGR	3	c12846.graph_c0, c17445.graph_c0
		c18037.graph_c0,
FPPS	2	c3736.graph_c0, c67.graph_c0
FPS	2	c67.graph_c1, c10924.graph_c0
PMK	2	c13594.graph_c0, c22627.graph_c0
SM	1	c25144.graph_c0
HMGS	5	c10205.graph_c0, c16013.graph_c0
		c26810.graph_c0, c20761.graph_c0
		c27696.graph_c0
DMD	1	c9873.graph_c0
FT	1	c15553.graph_c0

SS: squalene synthase, SE: squalene epoxidase, AACT: Acetyl-CoA acetyltransferase, HMGR: 3-hydroxy-3-methylglutaryl coenzyme A reductase, FPPS: Farnesyl pyrophosphate synthase, FPS: farnesyl-diphosphate synthase, PMK: Phospho mevalonate kinase, SM: squalene monooxygenase, HMGS: hydroxymethylglutaryl-CoA synthase, DMD: Diphospho mevalonate decarboxylase, FT: farnesyl transferase.

## Data Availability

The raw data required to reproduce these findings cannot be shared at this time as the data also forms part of an ongoing study. Enquiries about data availability should be directed to the authors.

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
