# Peer review of "Transcriptional Responses for Biosynthesis of Triterpenoids in Exogenous Inducers Treated Inonotus Hispidus Using RNA-Seq"

_molecules, 2022, doi:10.3390/molecules27238541_

Round 1
Reviewer 1 Report
The manuscript combined the chemistry and RNA-seq to study on the physiology and metabolism in this fungi (Inonotus hispidus), and the results seem interesting. However, I think the paper still need to be revised before accept.
1. In the abstract, line 13-17. If this is the previous study, you should abbreviate it.
2. In line 62-64, it is better to cite the reference in Fungi,
3. Line74-75, the author treat the sample with the JA, but the time of each treatment is not described clearly. In fact, the JA and oleic acid were added in the 6th day, what about the other days and the suitable time you determined for analysis?
4. Line 338-346 In the transcriptome part, the sample prepared detail should be added, it is important for your results.
5. In table 5 and figure 8 , the author discovered some high expressed unigenes. However, why not provided the experimental data to prove them?
Reviewer 2 Report
Huo et al. performed RNA-seq transcriptome analysis of fungus, Inonotus hispidus treated with MeJA and oleic acid to study the molecular background of triterpenoid accumulation in I.hispidus. They identified several genes associated with enzymes involved in the biosynthetic pathway of triterpenoids. Pathway analysis of differentially expressed genes showed enrichment of genes related to Ribosome and Translation.
The topic is interesting, and the results will help to extend research for identification of bioactive compounds.
Points to be addressed:
The authors state that for differential analysis of RNA-seq data they made comparisons: CK vs S1, CK vs S2, and CK vs S2. Assuming, that CK stands for “Control sample”, the authors should compare treatment against control (not the opposite), therefore the comparisons should be performed as: S1 vs CK, S2 vs CK, and S3 vs CK.
The numbers of genes in Venn diagrams of Figure 5a (up-regulated) and Figure 5c (down-regulated) don’t add up in the Figure 5d (labeled as All-DEG). In addition, the numbers of genes in Figure 5b don’t fully correspond with its description in the text.
The authors should include the list of 162 common genes into Supplementary Material Table.
Pathway analyses indicated term Ribosome with the greatest enrichment of differentially expressed genes. What are the enriched genes? The heatmap illustrating expression of these genes across all experimental conditions would greatly represent the subset of Ribosome genes involved in triterpenoid accumulation.
The authors talk about the results of ‘hierarchical clustering” in the text. However, there is no illustration of hierarchical clustering in the manuscript.
The names and versions of all software used for RNA-seq analysis need to be described.
